# Analogue viscous current flow near the onset of superconductivity

Koushik Ganesan* and Andrew Lucas†

*Department of Physics and Center for Theory of Quantum Matter,*
*University of Colorado, Boulder CO 80309, USA*

(Dated: April 7, 2023)

Spatially resolved transport in two-dimensional quantum materials can reveal dynamics which is invisible in conventional bulk transport measurements. We predict striking patterns in spatially inhomogeneous transport just above the critical temperature in two-dimensional superconducting thin films, where electrical current will appear to flow as if it were a viscous fluid obeying the Navier-Stokes equations. Compared to viscous electron fluids in ultrapure metals such as graphene, this analogue viscous vortex fluid can exhibit a far more tunable crossover, as a function of temperature, from Ohmic to non-local transport, with the latter arising on increasingly large length scales close to the critical temperature. Experiments using nitrogen vacancy center magnetometry, or transport through patterned thin films, could reveal this analogue viscous flow in a wide variety of materials.

## 1. INTRODUCTION

Despite transport being arguably the simplest possible experiment in solid-state physics, the electrical conductivity $\sigma$ can also be the most challenging quantity to predict theoretically, especially in strongly correlated systems. Over the coming decade, it will be increasingly possible to measure not only the bulk conductivity $\sigma$, but also a *wave number dependent* conductivity $\sigma(k)$, using either microscopically-etched devices [1], flows through channels of variable width [2–4], or local imaging probes. For example, nitrogen vacancy centers in diamond can be high-sensitive, nanometer-resolution magnetometers used to image electric current flow [5–7]. Scanning electron tunneling [8–10] could also be used to map local electric potential on similarly short distance scales. Clear predictions for what these future experiments will image across the plethora of discovered phases of quantum matter is a timely endeavor [11, 12].

Here, we study theoretically the spatially-resolved transport of a two-dimensional metallic system near the onset of superconductivity in the absence of external magnetic fields. We consider a system at temperature $T$ just above the critical temperature $T_c$, below which there is superconductivity and essentially no bulk resistivity. The experimental signature which gives superconductivity its name is that the bulk conductivity $\sigma(k = 0) \to \infty$ as $T \to T_c$ from above. Yet $\sigma(k > 0)$ is not likely to diverge as well: at finite $k$, the system will already appear ordered – nothing should seem singular at $T_c$. On phenomenological grounds, we can therefore conclude that

$$\sigma(k) \approx \frac{1}{a(T - T_c) + bk^c}, \quad \text{as } k \to 0. \tag{1}$$

where $a(0) = 0$ and $b, c > 0$ are approximately $T$-independent constants. Space-resolved transport should

be quite dramatic if $\sigma(k)$ stays finite at fixed $k \neq 0$ as $T \to T_c$.

We will argue that (1) indeed holds, with exponent $c = 2$. This dependence on $k$ is mathematically equivalent to what happens in a viscous electron fluid [5, 13–17]. Our theoretical results are grounded in the well-established physics of vortex dynamics near the BKT crossover in two-dimensional superconducting thin films [18–22], developed 40 years ago. The main result of this work is predicting the direct experimental signature of vorticity diffusion in non-local conductivities, which can in turn lead to analogue viscous current flows detectable in experiments. Our theoretical perspective follows closely more recent work [23, 24], and is suited for the strongly correlated dynamics at $T_c$. The length scale below which the response will appear viscous is set by the typical inter-vortex spacing, which diverges as $T \to T_c$. In a sufficiently clean device, it may be possible to image "viscous" flow patterns on large length scales just above $T_c$. Superconducting thin films are thus predicted to be an excellent platform to realize viscous current flow patterns, more robustly than in normal metals such as graphene or GaAs. Analyzing this analogue viscous flow can reveal fundamental information about the dynamics of strongly correlated electrons which is otherwise invisible in bulk resistance measurements.

## 2. SPATIALLY RESOLVED TRANSPORT

Before describing our derivation of (1), let us explain how this quantity can be (indirectly) probed in experiment. Due to the large speed of light $c$, one cannot simply shine light on a sample, since $\omega = ck$ is very large (in fact, it is then more appropriate to approximate $k = 0$ while $\omega \neq 0$).

We advocate the following strategy instead. Consider etching a constriction into a 2d material, as sketched in Figure 1. The blue region dictates the region where current cannot traverse. In experiments on graphene this is done by applying a bias voltage over the constriction, forming an effectively "hard wall" region where the Fermi

* koushik.ganesan@colorado.edu
† andrew.j.lucas@colorado.edu

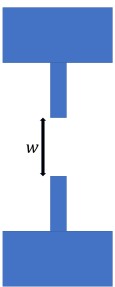

FIG. 1: Sketch of a constriction geometry of width $w$ with current flowing outside the blue region

energy is very different. When applying a constant uniform transverse electric field, the presence of these "hard walls" modifies the current flow pattern. A microscopically exact treatment of this problem has never been found. To compare with existing experimental data [6], therefore, a more phenomenological approach is needed. Consider the formal equations

$$J_i(\vec{x}) = \int \mathrm{d}^2\vec{x}' \sigma_{ij}(\vec{x}, \vec{x}') E_j(\vec{x}') \tag{2a}$$

$$\vec{E}(\vec{x}) = \vec{E}^{(0)} + \vec{E}_{\mathrm{ind.}}(\vec{x}) \tag{2b}$$

where $\vec{E}^{(0)}$ and $\vec{E}_{\mathrm{ind.}}$ are the background and induced electric fields respectively. The induced electric fields will vanish except inside of the forbidden regions where electrons cannot traverse. We thus require that outside the walls $\vec{E}_{\mathrm{ind.}} = 0$, while inside $\vec{J} = 0$. Making a final assumption that $\sigma(\vec{x}, \vec{x}') = \sigma(\vec{x} - \vec{x}')$ (the sample is otherwise approximately homogeneous), one can look for solutions to (2) consistent with these requirements. This prescription has been detailed at some length elsewhere [11, 12]. The key thing which has been found is that the current flow pattern is qualitatively modified by $\sigma(k)$, which is simply the Fourier transform of $\sigma(\vec{x} - \vec{x}')$:

$$\sigma(k)\left[\delta_{ij} - \frac{k_i k_j}{k^2}\right] = \sigma_{ij}(k). \tag{3}$$

Assuming translation invariance, the spatially dependent conductivity $\sigma_{ij}(\vec{x} - \vec{x}')$ can be formally defined as

$$\sigma_{ij}(\vec{x} - \vec{x}') = \lim_{\omega \to 0} \frac{1}{\omega} \mathrm{Im}\left(G^{\mathrm{R}}_{J_i(\vec{x})J_j(\vec{x}')}\right). \tag{4}$$

## 3. ANALOGUE VISCOUS FLOW IN SUPERCONDUCTING THIN FILMS

Consider a two-dimensional metal (or a thin film of thickness $d$ of an intrinsically three-dimensional metal) at temperature $T$ just above a superconducting transition temperature $T_{\mathrm{c}}$ (here $T_c$ corresponds to $T_{\mathrm{BKT}}$). In the 3d case, we require $d \ll \zeta_0$, where $\zeta_0$ is the 3d coherence length of the superconductor, which will be finite at the critical temperature of the 2d film. We take $T < T_{\mathrm{GL}}$, but $T > T_{\mathrm{c}}$. In this temperature range, vortices proliferate and destroy (quasi-)long-range order. As $T \to T_{\mathrm{c}}$, the diverging conductivity $\sigma(k = 0)$ is a consequence of the vanishing of "free" vortices (not bound in a vortex-antivortex pair) [25].

We acknowledge that the Aslamazov-Larkin (AL) theory of fluctuations [26–28], which strictly holds above $T_{\mathrm{GL}}$, can describe many experimental situations quite close to $T_{\mathrm{GL}}$ (which is in turn very close to $T_{\mathrm{c}}$). While we point out that the AL theory does not hold for $T - T_{\mathrm{c}} \ll T_{\mathrm{GL}} - T_{\mathrm{c}}$, there may be a narrow temperature window where the effects we predict are visible, and those of the AL theory are not. We leave a precise answer to this question to future work.

Strictly speaking, an essential difference between a superconductor and superfluid becomes important close to $T_{\mathrm{c}}$: as the supercurrents that circulate around a vortex generate a magnetic field, this leads to an effective potential energy between vortices that scales as $\sim r^{-1}$ instead of the usual $\sim \log r$ interaction of the superfluid [29]. However, in typical superconductors, the crossover between these power laws occurs at distances $\gtrsim 1$ mm [19, 30]. As this length scale is orders of magnitude larger than the mesoscopic samples we advocate looking for analogue viscosity in, we can safely neglect this correction. Henceforth, we can approximate that vortex dynamics will be the same as in a neutral superfluid.

To understand the implications of these long-lived vortices in transport, we observe that for $T - T_{\mathrm{c}} \ll T_{\mathrm{c}}$, there is a long-lived supercurrent $J_\phi$. In general, computing $\sigma(k)$ is incredibly challenging. However, because $J_\phi$ is long-lived, we expect $\sigma(k)$ to be dominated by the relaxation of $J_\phi$ [23, 24].

To explain more quantitatively, let $u_\phi \propto \nabla\phi$ be the superfluid velocity, which is thermodynamically conjugate to $J_\phi$. On general grounds [23, 31, 32],

$$J = \chi_{JJ_\phi} u_\phi + \cdots; \quad J_\phi = \chi_{J_\phi J_\phi} u_\phi + \cdots. \tag{5}$$

Using the memory matrix formalism [32, 33], one can show that

$$\sigma(k) = \frac{\chi^2_{JJ_\phi}(k)}{\chi_{J_\phi J_\phi}(k)\Gamma_{J_\phi J_\phi}(k)} + \cdots. \tag{6}$$

Here $\Gamma_{J_\phi J_\phi}$ is the relaxation rate of the supercurrent:

$$\partial_t \langle J_\phi \rangle = -\Gamma_{J_\phi J_\phi} \langle J_\phi \rangle. \tag{7}$$

Precise computations of any of the three terms in (6) are difficult near $T_{\mathrm{c}}$ in any microscopic model, but we can reliably estimate the scaling of each term, beginning with the susceptibilities. Importantly, we find that $\chi_{J_i J_\phi^j}$ and $\chi_{J_\phi^i J_\phi^j}$ are finite as $T \to T_{\mathrm{c}}$, and do not exhibit power law dependence in $k$. In the absence of rotational symmetry breaking, one can define them as follows: [23, 24]

$$\chi_{J_i J_\phi^j} \approx q\rho_{\mathrm{s}}\delta_{ij}, \quad \chi_{J_\phi^i J_\phi^j} \approx m\rho_{\mathrm{s}}\delta_{ij}. \tag{8}$$

Here the superfluid boson degree of freedom has mass $m$ and charge $q$, and $\rho_s$ is the local bare superfluid density outside of vortex cores, which is finite near $T_c$ [19]: see the appendices for further discussions. For simplicity writing $\Gamma_{J_\phi^y J_\phi^y}(k) = \Gamma(k)$, we conclude that

$$\sigma(k) \approx \frac{q^2 \rho_s}{m \Gamma(k)}. \tag{9}$$

The non-trivial calculation is thus of $\Gamma(k)$.

$\Gamma(k)$ is where the vortex physics highlighted earlier becomes important. The superfluid velocity $u_\phi = (\hbar/m)\nabla\phi$ is the gradient of a phase, a U(1) order parameter which exhibits point-like defects called vortices in two dimensions. The stable vortices have phase $\phi$ which winds by $\pm 2\pi$ around a point: let us denote the density of free positive circulation and negative circulation vortices with $n_f^+$ and $n_f^-$ respectively. The density of free vortices is given by $n_f = n_f^+ + n_f^- \sim \zeta^{-2}$, where [20, 22]

$$\zeta = \zeta_0 \exp\left[ b\sqrt{\frac{T_c}{T - T_c}} \right] \tag{10}$$

with $b$ a material-dependent constant (usually $b \sim 0.1$ [34]). In contrast, the signed vortex density is

$$n_v = n_f^+ - n_f^- = \nabla \times \nabla\phi = \frac{m}{\hbar}\nabla \times u_\phi. \tag{11}$$

Since spatial variations are aligned along $x$,

$$\mathrm{i}k J_\phi^y(k) \sim \mathrm{i}k u_\phi^y \sim n_v(k), \tag{12}$$

and thus we can deduce the leading order behavior in $\Gamma(k)$ by calculating the signed vortex decay rate.

Just above $T_c$ the vortices form an analogue "Coulomb gas" [22] of free charges (vortices); the superfluid velocity $\sim 1/r$ is orthogonal to an effective electric field $\sim 1/r$ that would be generated by the charges. $\Gamma(k)$ can thus be deduced via the relaxation of externally imposed charges in a two-dimensional plasma. If there were no free vortices ($n_f = 0$, or $T = T_c$), then we expect diffusive relaxation: $\Gamma(k) \approx D_0 k^2$ on length scales $k^{-1} \gg \zeta_0$, simply because this is the generic hydrodynamics of a conserved quantity [20, 21]. However, due to long-range interactions between vortices, $\Gamma(0) > 0$ is finite if $n_f > 0$: the mechanism is mathematically identical to the finite time decay of free charges in Maxwell's equations with an Ohmic current ($J \propto E$). Hence for a constant $c_0$ (see SM for details),

$$\Gamma(k) \approx D_0(c_0 n_f + k^2 + \cdots). \tag{13}$$

$\cdots$ indicates higher-order terms suppressed by additional powers of $\zeta_0 k$. Combining (6), (8) and (13), we find

$$\sigma(k) \approx \frac{q^2 \rho_s}{m} \cdot \frac{1}{D_0(c_0 n_f + k^2)}. \tag{14}$$

This is the main result of our paper. Our claim is that (14) captures the generic scaling of $\sigma(k)$ both when

$k \ll \sqrt{n_f}$, and when $\sqrt{n_f} \ll k \ll 1/\zeta_0$, independently of the microscopic details. This is because in a generic superfluid, the only parametrically long-lived mode near $T_c$ is the supercurrent. If we take $k \to 0$ in (14), we have just computed the bulk conductivity of the metal just above $T_c$. The divergence in conductivity as one approaches superconductivity follows from the vanishing of free vortex density, which is simply the Bardeen-Schrieffer "flux flow conductivity" $\sigma \sim 1/n_f$ [25], which has been experimentally observed [35–37]. In the limit of finite $k$, (14) implies $\sigma(k) \sim k^{-2}$. This can intuitively be thought of as a consequence of vortex diffusion. In the SM, we discuss why there are indeed no anomalous corrections to diffusion (a point first made in [21]), and that the diffusion constant itself (while being the physical diffusion constant for vorticity relaxation) is dominated by the response of tightly bound vortex pairs. Importantly, (14) applies to both conventional and unconventional superconductors, as the arguments for the form of $\Gamma(k)$ make no reference to the conventional BCS theory of superconductivity.

Remarkably, the exact same mathematical structure as (14) also arises if one models the electrons as a viscous fluid [13–17]: solving the Navier-Stokes equations in the presence of impurities, one finds [11, 12]

$$\sigma(k) \approx \frac{\rho_0^2}{\eta(k^2 + \ell^{-2})}, \tag{15}$$

where $\rho_0$ is the normal charge density and $\eta$ is the shear viscosity, and the scattering rate off of impurities, which relaxes momentum, is proportional to $\ell^{-2}$. The mathematical analogy between (14) and (15) makes precise our claim that one can look for analogue viscous flows just above $T_c$. We emphasize however that in general, a viscous electron fluid only arises when there is approximate momentum conservation in electronic collisions, which is a very rare criterion (most metals are quity dirty). In contrast, any metal with a superconducting transition will eventually reach a regime very close to $T_c$ where $n_f \to 0$. So it should be *easier* to see "viscous flows" near the onset of superconductivity, than in a genuinely viscous electron fluid.

As we detail in Appendix B, our microscopic argument for the scaling $k^{-2}$ assumes that we are exactly at $T = T_c$; it is possible that for $T < T_c$, corrections to this scaling arise. While our microscopic argument does not suggest that the exponents of the equilibrium inter-vortex distribution modify the scaling exponent in $k^{-2}$, a more detailed analysis could be worthwhile. Even if such corrections do eventually emerge far from the critical temperature, from an experimental point of view, observing these corrections could be challenging, as even observing the scaling given in (14) will require a carefully designed experiment!

## 4. COMPARISON TO PREVIOUS STUDIES

There have been several older studies looking at viscosity in the context of vortex liquids in superconductors and we now compare our findings to earlier work. The first studies of $\sigma(k)$ [38–43] focused on the dynamics of melted Abrikosov flux lattices in large magnetic fields, whereas our work focuses on the zero field limit. Moreover, what we call "analogue viscosity" is *not the same* as the vortex viscosity identified in these references, which was taken to be the $k^2$-coefficient of a Taylor expansion of $\sigma(k)$. Therefore, they claimed that $\eta_{\text{vortex}} \sim D_0^{-1} n_{\text{f}}^{-2}$. Comparing (14) and (15) we see that $\eta_{\text{analogue}} \sim D_0 n_{\text{f}}^0$. Finally, [42] argues that in this regime, $\sigma(k)$ is a non-monotonic function which *increases* at small $k$, not at all like (14). Interestingly nevertheless, $\sigma \sim k^{-2}$ was found in [41] in the limit $T \gg T_{\text{GL}}$, although this calculation could not explain the origin of $\sigma(k) \sim k^{-2}$ near $T_{\text{c}}$ from vortex physics, which is a non-perturbatively small correction at high $T$. In our calculation we implicitly assumed a strongly interacting vortex liquid near $T_{\text{c}}$.

The physics responsible for analogue viscosity was first identified in an even older literature [20, 21] on vortex diffusion in superfluid thin films. The contribution of this work is (14): vorticity diffusion should be readily measurable via non-local conductivity $\sigma(k)$. In other settings, authors have found logarithmic corrections to $D_0$ [44, 45] which arise due to long-range interactions between vortices. However, as argued in [21], we do not expect this effect to arise near the onset of superconductivity: see Appendix. In any case, the existence or not of any logarithmic corrections in $\sigma(k)^{-1}$ will be quite hard to detect in experiment.

It has also been recently argued [46] that the true shear viscosity $\eta$ is strongly $T$-dependent close to $T_{\text{GL}}$. However, we emphasize that our mechanism is qualitatively distinct: the perturbative calculation of [46] which suggests viscous flow in superconducting metals is in the regime $T > T_{\text{GL}}$, whereas our calculation holds when $T < T_{\text{GL}}$.

## 5. EXPERIMENTAL IMPLICATIONS

The link between vortex diffusion and non-local conductivity identified in this work has serious experimental implications, as we now discuss. Due to the large speed of light, optical probes readily measure $\sigma(k = 0, \omega \neq 0)$, rather than $\sigma(k \neq 0, \omega = 0)$; indeed, 40 years ago, most experimental works focused on finite $\omega$ (not $k$) response for this reason. However, we have emphasized here that $\sigma(k)$ is a direct window into vortex diffusion, an effect which is readily seen in $\sigma(\omega)$. Perhaps the most direct way to measure the latter is by using local magnetometry. One can either measure current fluctuations (which give $\sigma(k)$ due to the fluctuation-dissipation theorem [47]), or simply image current flow patterns [5–7] and deduce the resulting $\sigma(k)$ [11, 12]. Imaging techniques such as

nitrogen-vacancy center magnetometry have been successfully demonstrated at 6 K [48], meaning that this technique is capable of testing our predictions in conventional superconducting thin films. This imaging can be done at $\sim 50$ nm, but even assuming that we can only probe much larger length scales $L \sim 1~\mu$m, we could detect a crossover from Ohmic to viscous flow patterns when $L \sim \zeta$. After all, the free vortex spacing $\zeta$ diverges near $T_{\text{c}}$. Using estimates $b \approx 0.1$ and $\zeta_0 \approx 8$ nm from NbN thin films [34, 49] (which we expect are reasonable order-of-magnitude estimates for most conventional superconducting thin films), one must be within about 0.1% of $T_{\text{c}} \sim 7.7$ K in order to find $L \sim 1~\mu$m. At this length scale the resistivity will be reduced by the factor of $(\zeta_0/\zeta)^2 \sim 10^{-4}$, which has been resolved in experiment. Hence we expect such precision in $T$ is achievable, albeit may prove challenging in magnetometry experiments (where a rise in temperature due to electronic heating must be balanced against the current density which is detectable by the magnetometer itself.

If probing $\sigma(k)$ directly is not possible, one may alternatively engineer devices with narrow constrictions through which current is forced to flow. Mirroring known results for viscous electron fluids [16, 17], we estimate the resistance $R$ of the device sketched in Figure 1 as

$$R = \frac{32mD_0}{\pi q^2 \rho_{\text{s}}} \left[ c' c_0 n_{\text{f}} + \frac{1}{w^2} \right] \qquad (16)$$

where $c' \sim 1$ will depend on the geometry of the device and may exhibit some dependence on ratios of $w$ to the size of the device as a whole [17]. However, once $w \ll \zeta$, the dominant contribution to $R$ will come from the $1/w^2$ term, heralding the analogue viscous flow. In Figure 2 we sketch how $R(T)$ should look as a function of $T$ close to $T_{\text{c}}$. This prediction assumes that generic boundary conditions hold on the current, and that the current does not exactly follow the unique curl-free flow pattern (see Appendix). Since almost three decades ago non-local transport was observed on $\mu$m scales in YBCO [50] in a $\sim 5$ T field, we expect that currents must obey boundary conditions conducive to the non-local transport described above, at least in some materials.

Patterning such constrictions into a thin film superconductor can be done via focused ion beam etching [3]. Alternatively, local gates above the device can be used to imprint a constriction geometry, as is readily done in graphene-based devices [6]. Assuming generic boundary conditions, imaging current flows through these geometries with magnetometers will lead to qualitatively different flow patterns in Ohmic regimes ($\sigma(k) \sim k^0$) vs. non-local regimes (e.g. $\sigma(k) \sim k^{-2}$), and is a more compelling experimental probe of non-local transport than a conductance measurement by itself.

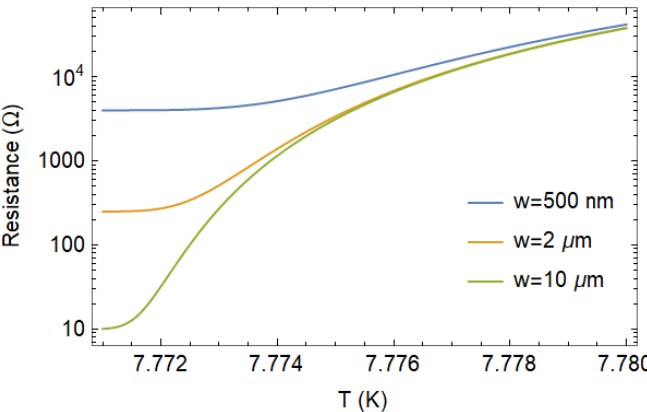

FIG. 2: Resistance for constrictions of widths 0.5, 2 and 10 $\mu$m for a 3 nm NbN film near the transition temperature $T_c = 7.7$ K using estimated parameters [34, 49].

## 6. DISCUSSION AND CONCLUSION

A natural candidate material to test our theory is MoGe [51], which exhibits an "anomalous" metal which is likely a failed superconductor [23]. A rather different application of this technique could be to gain further insight into the "strange metal" phase observed in both twisted magic-angle bilayer graphene [52–54], and thin films of a cuprate superconductor [55]. Understanding the role played by vortices and phase fluctuations in bad metals could help deduce the origin $T$-linear resistivity and provide a further check for universality in dynamics from one material to the next. Evidence for such a conjecture thus far relies on the universality of "Planckian" scattering rates in bulk transport; imaging local currents will provide an important check into the interplay of non-quasiparticle physics with the onset of ordering. In particular, Planckian-limited scattering rates might be observed in the diffusion constant $D_0$, even as the bulk resistivity (which is approaching zero near $T_c$, and is clearly well below the Planckian bound conjectured in [56]. Therefore, studying spatially resolved transport could reveal valuable insight about potential universal properties of strange metals near $T_c$. It is predicted [23, 25] that near $T_c$, the vortex diffusion constant $D_0 \sim \rho_n$, the electrical resistivity of the normal fluid component. Hence measuring the strength of "analogue viscous flow", which directly determines $D_0$, can lead to a test of whether the Planckian scattering responsible for $\rho \sim k_B T/\hbar$ well above $T_c$, continues to hold at $T_c$. Additionally, in ordinary superconducting thin films it is expected that $D_0 \gtrsim \hbar/m$ [20], and it would be interesting to learn experimentally whether the alternative bound $D_0 \gtrsim v^2 \hbar/k_B T$ proposed in [56] holds near the superconducting transition of a strange metal.

One challenge for realizing this non-local transport over parametrically large scales in a real material may be inhomogeneities – not because they can directly relax the

"viscous" mode as in a conventional electron fluid [13], but because they could cause $T_c$ to substantially vary throughout the device [57]. This inhomogeneity could put a fundamental limit on the regime of validity of our theory, which may not capture dynamics across superconducting islands. Testing our predictions in materials where the onset of superconductivity appears quite sharp, as measured in device resistance $R(T)$ as a function of temperature, may be a practical way to minimize these effects [30]. On the other hand, dirty superconductors have a broader crossover in the resistivity $\rho(T)$, suggesting that less temperature control will be necessary to access regimes close to $T_c$ where vortex physics can dominate. Future experiments will help to deduce the criteria and materials where non-local transport is most readily observed in a superconducting thin film.

### ACKNOWLEDGEMENTS

We thank Luca Delacretaz, Dan Dessau, Sean Hartnoll, and especially David Huse and Leo Radzihovsky, for helpful discussions. This work was supported by the Alfred P. Sloan Foundation through Grant FG-2020-13795 (AL), and by the Gordon and Betty Moore Foundation's EPiQS Initiative via Grant GBMF10279 (KG, AL).

### Appendix A: Thermodynamics

To motivate (8), consider a microscopic Ginzburg-Landau theory of a superconductor with Lagrangian

$$\mathcal{L} = i\hbar\bar{\psi}\partial_t\psi - \frac{|\hbar\nabla\psi - iq\vec{A}\psi|^2}{2m} - V(|\psi|) + \cdots \quad (A1)$$

Here $m$ is the mass of the condensing boson (which for conventional superconductors is the Cooper pair) and $q$ its charge. Writing $\psi = \sqrt{\rho}e^{i\phi}$, the supercurrent is

$$\vec{J}_\phi = \rho(m\vec{u}_\phi - q\vec{A}) \quad (A2)$$

with $u_\phi = \frac{\hbar}{m}\nabla\phi$. Far from a vortex core, we have $\rho \approx \rho_s$, the bare superfluid density. Hence we obtain

$$\chi_{J_\phi^i J^j}(k=0) = -\frac{\partial J_\phi^i}{\partial A_j} = q\rho_s\delta_{ij}, \quad (A3)$$

and we do not expect this susceptibility to exhibit strong $k$-dependence.

Next we turn to $\chi_{J_\phi^y J_\phi^y}(k)$. This is a little more subtle because, as emphasized in the main text, $J_\phi^y$ is closely related to vortex physics, which can seem singular as $n_f \to 0$. We invoke the Coulomb gas analogy to a charged plasma, where we can calculate the vortex susceptibility $\chi_{n_v n_v}$ using the following identity relating (analogue) charge susceptibility to permittivity [58]:

$$\frac{1}{\epsilon(k)} = \frac{1}{\epsilon_0} - \frac{\chi_{n_v n_v}(k)}{k^2\epsilon_0}. \quad (A4)$$

where $\epsilon_0$ is the bare dielectric constant without including free vortices and the polarization of bound vortices (which we keep just for clarity, though it may as well be 1). One expects that for $T \approx T_c$ (for some constant $\kappa > 1$ and O(1) constant $b_0$) [22]

$$\frac{1}{\epsilon(k)} \approx \frac{1}{\kappa\epsilon_0}\left[\frac{k^2}{k^2 + b_0 n_f} + ak^2 + \cdots\right], \qquad \text{(A5)}$$

where $a \sim \zeta_0^2$ is a non-singular term. The first term is extremely singular and corresponds to the Poisson-Boltzmann screening of point charges whenever $n_f > 0$, since the combined potential $V(k) \sim n_v(k)/k^2\epsilon(k)$; the second term is regular and just corresponds to microscopic scale corrections to the Coulomb gas analogy to point charges. Combining (A4) and (A5) and neglecting the $a$-term, we find

$$\chi_{n_v n_v}(k) = k^2 \frac{k^2(1 - \kappa^{-1}) + b_0 n_f}{k^2 + b_0 n_f}. \qquad \text{(A6)}$$

Combining (A6) with (11) and (12) we deduce that

$$\chi_{J_\phi^y J_\phi^y} \approx m\rho_s \frac{k^2(1 - \kappa^{-1}) + b_0 n_f}{k^2 + b_0 n_f}. \qquad \text{(A7)}$$

If $\kappa \gg 1$, we can essentially treat this susceptibility as a constant; if $\kappa \gtrsim 1$, then this susceptibility could change by some O(1) factor; however this would modify neither limit of our main result (14).

## Appendix B: Vortex decay time

Now we estimate $\Gamma(k)$ with the memory matrix formalism [31, 32]. Observe that $n_v(k)$ is (generically) the unique parametrically long-lived pseudoscalar operator (at wave number $k$) in the system; therefore, we can write

$$\partial_t \langle n_v(k) \rangle \approx -\frac{M(k)}{\chi_{n_v n_v}(k)} \langle n_v(k) \rangle, \qquad \text{(B1)}$$

where the memory matrix

$$M(k) \approx \lim_{\omega \to 0} \frac{1}{\omega} \text{Im}\left[G_{\dot{n}_v \dot{n}_v}^R(k, \omega)\right], \qquad \text{(B2)}$$

where $\dot{n}_v = i[H, n_v]$ is the operator corresponding to the rate of change of vortex density. We may write

$$\dot{n}_v(k) = ik \cdot J_v(k), \qquad \text{(B3)}$$

where $J_v(k)$ is the vortex current operator. In a semiclassical limit with large vortices,

$$J_v(k) = \sum_a \Gamma_a \dot{x}_a e^{ik \cdot x_a} \qquad \text{(B4)}$$

where $a$ indexes the vortices in the system with orientations $\Gamma_a = \pm 1$.

To proceed farther and evaluate the spectral weight (B2) we need to make a few more assumptions. For simplicity let us begin working exactly at $T = T_c$. Following the Coulomb gas literature, at this point all vortices are "bound" in pairs; letting $\zeta_0$ denote the superfluid healing length (which is finite, and corresponds to roughly the radius of the vortex core), one finds [21] that the probability density $p(r)$ that a vortex pair is separated by distance $r$ is

$$p(r) \sim \frac{\zeta_0^2}{r^3 \log^2(r/\zeta_0)}. \qquad \text{(B5)}$$

Let us further assume the dominant contribution to the spectral weight arises from interactions between two vortices in a vortex pair (possibly screened via other more tightly bound pairs, but this is accounted for by renormalized coefficients [22]); in this case we estimate that

$$M(k) \approx \int_{\zeta_0}^{\infty} dr \frac{\zeta_0^2}{r^3 \log^2(r/\zeta_0)} K(r), \qquad \text{(B6)}$$

where

$$K \sim k^2 \int_{-\infty}^{\infty} dt \left\langle \left(\dot{x}_+(t)e^{ik \cdot x_+(t)} - \dot{x}_-(t)e^{ik \cdot x_-(t)}\right) \cdot \right.$$
$$\left. \left(\dot{x}_+(0)e^{-ik \cdot x_+(0)} - \dot{x}_-(0)e^{-ik \cdot x_-(0)}\right) \right\rangle \qquad \text{(B7)}$$

with $x_\pm(t)$ denoting the position of the positive/negative vortex in the pair, obeying $|x_+ - x_-| = r$; the average $\langle \cdots \rangle$ This formula is rather imprecise, as a pair will tend to dynamically change its value of $r$ over time, but intuitively our approximation is to understand the contribution of each bound pair on its own to $M(k)$.

In the absence of thermal fluctuations *and* dissipation (both coming due to interaction with the normal fluid), a bound pair will simply propagate in a straight line, perpendicular to the orientation of the vortex dipole. Orienting the dipole for simplicity by placing the $\pm$ vortex at $(0, \pm r/2)$, we find this bound pair's contribution to

$$k \cdot J_v(k) \sim \frac{k_x \hbar}{mr} \sin(\frac{k_y r}{2}). \qquad \text{(B8)}$$

However, since the isolated pair will move in a straight line forever, it's contribution to $K(r)$ would be $\infty$ due to the integral over $t$; thus we need to also incorporate dissipative effects. In the presence of dissipation (but not thermal noise), a bound vortex pair will deterministically annihilate in a time $\tau(r) \sim \eta r^2$ [59]. Accounting for this finite lifetime, and assuming that interactions with other pairs might only *decrease* the scaling of this lifetime (e.g. if a tightly bound and weakly bound positive vortex "swap roles" during a close pass, we expect the direction of the bound pair drift to become a bit randomized) we estimate that

$$K(r) \lesssim \frac{k^4 \hbar^2}{m^2} \tau(r) \left(\frac{\sin(kr)}{kr}\right)^2 \sim \frac{k^4 \eta \hbar^2}{m^2} r^2 \qquad \text{(B9)}$$

where we have taken the $r \to \zeta_0 \ll k^{-1}$ limit since, as we will see immediately, the smallest vortex pairs dominate the spectral weight. Plugging in (B9) into (B6) we obtain

$$M(k) \approx \frac{k^4 \eta \hbar^2}{m^2} \int_{\zeta_0}^{\infty} \frac{\mathrm{d}r}{r \log^2(r/\zeta_0)} \sim \frac{k^4 \eta \hbar^2}{m^2}, \qquad \text{(B10)}$$

since the integral over $r$ is convergent. (Although the convergence is slow, accounting for $\sin(kr)/kr$ corrections would give at best $k^4/\log(k\zeta_0)$, which is subleading compared to the contribution from $r \sim \zeta_0$.). Hence

$$\Gamma(k) = \frac{M(k)}{\chi_{n_{\mathrm{v}} n_{\mathrm{v}}}} \sim k^2, \qquad \text{(B11)}$$

with no logarithmic corrections despite the power-law nature of the interactions in the system. Indeed, observe that even if our estimate of $\tau(r)$ was too large, because the integral over $r$ in $M(k)$ is dominated at small $r$ not large $r$, a modified $\tau$ can only change $M(k)$ by an O(1) constant, but not lead to any anomalous $k$-scaling in the answer at leading order. [21] has also confirmed that perturbative corrections to this argument due to other more tightly bound pairs, which serve primarily to renormalize the effective permittivity of the vortex gas, do not lead to singular-in-$k$ corrections to these integrals.

We believe that for $T < T_{\mathrm{c}}$, the primary difference will be that the probability density $p(r)$ decays faster with $r$ than given in (B5) [22]. This would most likely suppressing subleading $k$-dependent corrections but otherwise, we do not expect it to alter our primary result.

There is one important subtlety overlooked thus far in the argument, which we must address. Naively, the dominant contribution to $k \cdot J_{\mathrm{v}}(k)$ arises due to the drift of vortex pairs together (or apart), which could contribute (in the absence of noise)

$$k \cdot J_{\mathrm{v}}(k) \sim -\frac{\eta k_y \hbar}{mr} \cos\left(\frac{k_y r}{2}\right), \qquad \text{(B12)}$$

which seems to dominate the scaling found in (B8). Indeed, at time $t = 0$, this "dissipative" part of the vortex current will dominate $\langle J_v(k, t \to 0) J_v(k, 0)\rangle$. However, we claim that when *integrating* the correlator over all time $t$, as in (B7), this contribution will be small (at least for the tightly bound pairs). Indeed, suppose that the superfluid consisted of a single bound pair, which popped in and out of existence (mainly staying tightly bound). Applying a weak "analogue electric field" (supercurrent), by the fluctuation-dissipation theorem, the spectral weight $K(r)$ will pick up a contribution proportional to the average contribution of this pair to $J_{\mathrm{v}}(k)$ as $t \to \infty$, even when averaging over orientations of the vortex dipole. This will vanish: the pair is bound and the electric field is not strong enough to overcome the "activation barrier" to pull the pair apart. Moreover, the actual drift of the vortex dipole in the background velocity field does not on average contribute to $J_{\mathrm{v}}(k)$, as

it flips sign when exchanging the positive/negative vortex (and each configuration is equally likely). Thus $K(r)$ *cannot* pick up any contributions from the vortex current associated with fluctuations in vortex distance, unless the vortex pair is widely separated, where the electric field *can* unbind it. Yet similar to the previous argument, the contributions to $M(k)$ from such weakly bound pairs are suppressed by a power of $1/\log(k\zeta_0)$.

For $T > T_{\mathrm{c}}$, we expect that

$$M(k) \sim k^2 \left(k^2 + c n_{\mathrm{f}}\right) \qquad \text{(B13)}$$

for O(1) constant $c$. In our argumentation, this happens because the bound dipole distribution $p(r)$ will truncate at $r \gtrsim \zeta \sim n_{\mathrm{f}}^{-1/2}$. Beyond this point, the contribution of unbound vortex motion to $k \cdot J_{\mathrm{v}}$ in (B8) will not come with an extra factor of $\sin(kr)$; thus we will find an additional contribution, of relative magnitude $(kr)^{-2} \sim (k\zeta)^{-2} \sim n_{\mathrm{f}}/k^2$ relative to the bound pair contribution; this leads to (B13) and thus to (13).

There is a somewhat curious observation that we can make. We have just argued that $M(k)$ is dominated by tightly bound pairs; however, from the equation of motion for $n_{\mathrm{v}}(k)$, we *also* conclude that the effective diffusion constant of one isolated vortex (also known to be finite [21]), must be given by the diffusion constant arising due to the dynamics of tightly bound pairs. Perhaps the microscopic mechanism for this vortex's diffusion is repeated interactions with annihilating/forming vortex pairs nearby, which could nicely explain why the tightly bound pairs appear to control the "macroscopically observable" diffusion constant of net vorticity.

Let us also briefly connect our somewhat microscopic arguments for $\Gamma(k)$ to the more recent works [23]. They wrote down a constitutive relation for the vortex current:

$$J_{\mathrm{v}}^i = C\epsilon^{ij} u_\phi^j - D_0 \partial^i n_{\mathrm{v}} + \cdots, \qquad \text{(B14)}$$

where $C \propto D_0 n_{\mathrm{f}}$. Together with the constitutive relation for the supercurrent (here $\hbar = 1$)

$$m\partial_t u_\phi^i = E^i - \partial^i \mu + 2\pi \epsilon^{ij} J_{\mathrm{v}}^j + \nu \partial^i \partial^j u_\phi^j + \cdots, \quad \text{(B15)}$$

and the relation $n_{\mathrm{v}} \propto \epsilon^{ij} \partial^i u_\phi^j$, one can immediately solve (B15) and find that $D_0[c n_{\mathrm{f}} + k^2] u_\phi^y(k) \sim E(k)$, which leads to (14). In [23], they did not compute $\sigma(k)$ and focused instead on $\sigma(\omega)$. From this perspective, our argument for $\sigma(k) \sim k^{-2}$ is clearly immediate; however, the subtle point is whether or not the vortex interactions, which are necessarily very long ranged, break the gradient expansion of hydrodynamics. In agreement with other authors, we have argued that this does not happen.

## Appendix C: Boundary conditions

What we have predicted is that any time transverse current flows near $T_{\mathrm{c}}$, one will see non-local conductivity due to vortex physics. Somewhat annoyingly, it is

possible to find a purely longitudinal flow in any two-dimensional geometry for any fixed boundary conditions on the current. Expressing such a current flow pattern as $\vec{J} = -\sigma_0 \nabla V$ (we have chosen suggestive notation, whereby $\sigma_0$ represents $\sigma(k = 0)$ (the diverging bulk conductivity near $T_c$) and $V$ represents a characteristic voltage profile which would be consistent with Ohmic flow), the uniqueness of solutions to $\nabla^2 V = 0$ with Neumann boundary conditions implies that one can find a current flow pattern obeying $\nabla \times J = 0$. It is not clear that on this flow pattern vortex physics must be relevant. However, if current flow patterns deviate at all from this special profile, then viscous effects must contribute to the conductance to some degree. Since experiments [50] have seen some evidence for non-local transport in vortex liquids in superconducting YBCO, we presume that at least in some materials, the boundary conditions are not consistent with purely longitudinal flow.

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
