# Peer review of "Analogue viscous current flow near the onset of superconductivity"

_SciPost Physics_

## Round 2 · Referee Report · Anonymous · 2022-10-16

Report

I got stuck at: "...we apply a small time-independent electric field E⃗ = Ey(k)e^ikx yˆ,...". If this is the only E-field present, then, by Faraday (I think), we have a steadily changing B-field, and the superconductor should "care" about that B-field. So it is not clear what the set-up is. We could make a steady-state like this with a strip and currents flowing across the strip from edge to edge (and nonuniform along the infinite direction of the strip). If that is what is intended it should be made explicit.

---

## Round 2 · Referee Report · Anonymous · 2022-10-20

Report

Like the other anonymous reviewer, I cannot go beyond Equation 1. How can one apply a spatially modulated electric field to a metal? The authors propose to apply an electric field with spatial periodicity. I am not aware of any previous implementation of this. Do the authors believe that this is feasible by current technology, or is it just a thought experiment?

---

## Round 3 · Referee Report · Anonymous (Referee 1) · 2022-12-27

Report

An issue of terminology: "non-Ohmic" usually means current nonlinear in voltage (and vice versa). In this paper it appears that non-Ohmic is being used to mean linear but non-local. This is potentially misleading and should be stated much more clearly, since this is quite non-standard usage. I think it would be best to not use "non-Ohmic" to mean anything other than nonlinear, thus following standard usage. Non-local is probably a clearer and less ambiguous term in this context.

Since this paper is a discussion of systems near the Kosterlitz-Thouless transition, there should be a more careful and thorough discussion of the contributions of weakly bound vortex-antivortex pairs. The conductivity at k should, roughly, "see" weakly-bound pairs with spacings more than 1/k as effectively free vortices. Since the pairs interact logarithmically with distance, this should bring in power laws with continuously variable exponents as one varies the temperature, as is standard in Kosterlitz-Thouless physics. Ref. 20 works this out for uniform currents (k=0) as a function of the frequency. It seems that here one should do the analogous calculation, using the Kosterlitz-Thouless RG understanding of these systems, for zero frequency as a function of the wavenumber k. Perhaps an even simpler geometry to think about for theoretical purposes is a Corbino geometry, where a weak current (in linear response) is injected (from out of the plane) at a point (or a small patch) and flows in the plane out to infinity isotropically. There will be extra dissipation near the source, due to weakly bound vortex-antivortex pairs.

---

## Round 3 · Referee Report · Anonymous (Referee 2) · 2023-1-26

Strengths

This is a pioneer paper about hydrodynamics in superconductors with rather specific predications, which can be experimentally verified.

Weaknesses

Neglect of real-world complications and in particular paraconductivity just above Tc.

Report

The paper argues that in a 2D superconductor in a narrow temperature window above the BKT critical temperature and below the GL critical temperature, the conductivity should depend on wave-vector because of the viscosity due to the presence of short-lived vortices. The main result of the paper is equation 14, which implies that DC conductivity at finite wave-vector should be smaller than DC conductivity at k=0. The authors argue that this can be tested by measuring the electrical resistivity of a 2D superconductor in a constricted geometry.

I think this is a nice contribution to the ongoing quest for finding different contexts for the observation of viscous electronic flow. The theoretical prediction looks easy to verify. In Figure 2, the authors show what the temperature dependence of the resistance of such constricted superconductors should be. However, they do not compare them with the actual resistivity curves reported in ref. 32 and 47 on much wider samples. Now, samples with a thickness of ~nm and a width of ~mm, show rounded transitions above Tc because of the well-known contribution of short-lived Cooper pairs (This is known as Aslamazov-Larkin fluctuations) . This contribution, the famous paraconductivity, is prominent up to at least 10 percent of Tc. In contrast, the effect discussed in this paper is restricted to a much narrower temperature range (~0.001 Tc). Just by comparing the evolution of the three theoretical curves for 0.5, 2 and 10 microns, one can see that it is unlikely to find what is seen in ref. 47 for a 1000 micron sample (such as S197 in their Table 1 and Figure 1).
All this suggests that the paper may be a nice academic contribution, but less directly connected to experimental reality than it is claimed.

  • validity: -
  • significance: -
  • originality: good
  • clarity: high
  • formatting: -
  • grammar: -

Author:  Andrew Lucas  on 2023-04-20  [id 3602]

(in reply to Report 2 on 2023-01-26)

Please see attached our response to the referees!

Attachment:

response_to_referee___superconductivity.pdf

---

## Round 3 · Author Response

The referees were confused about our remark about applying a spatially-varying electric field, so we have modified the text accordingly, including a new Section 2 that summarizes our approach. We have removed the reference to a spatially varying electric field which does not need to be applied as an external probe field in experiment. Since in experiment we do not propose applying time-dependent electromagnetic fields, we hope this clarifies the referees' confusion.

The notion of sigma(k), the wave number dependent conductivity, is not new to this paper. It is an established linear response coefficient that can be computed independently of any electric/magnetic fields that one applies -- it is simply a two-point function of the current operator. So it is perfectly logical as a theory question to try and calculate this object. Eq. (17) of 1708.02376 is a paper by other authors that employs similar ideas and explicitly calculates a scale dependent conductivity sigma(k).

More important is the question of why one should care what this calculation gives you. In the majority of the paper, and in part of Section 2, we address this point. Previous experiments (in particular Refs. [6,17]) have given affirmative evidence for the theoretical expectation that the wave-number-dependent sigma(k) can be used to quantitatively predict the inhomogeneous flow of current through an etched device (a la the constriction geometry in Figure 1). The main point of this paper is thus to predict what sigma(k) would be just above Tc in a superconducting 2d thin film, on experimental length scales of a few microns or smaller, and to discuss how this might be observed in experiment.

We hope this clarifies the confusion of the referees.

---

## Round 3 · List of Changes

See response: added new Section 2 and removed comments on applying a spatially varying electric field.

---

## Round 4 · Referee Report · Anonymous (Referee 1) · 2023-4-29

Report

I still do not understand the basic set-up that this paper is trying to discuss.

They use the pattern shown in Fig. 1. As I said earlier, I suspect that this would be cleaner to discuss theoretically for the Corbino geometry, where the fields and currents are rotationally symmetric about a central circle where the DC current is being injected (and flowing radially out to infinity).

I did not find a statement of what is meant by E^(0) in eqn. 2(b). A few lines above a "constant uniform transverse electric field" is mentioned, and looking at ref 12 it seems that this might be E^(0). As is mentioned in ref 12, this "external" field will induce some pattern of space charge in the (super)conducting film. If the material is a good superconductor close to its transition, the induced field should mostly cancel the background field within the film, making the field inside the superconductor much smaller, and concentrating the electric field within the barrier. But this is not what is said below eqn 2(b). Why would the induced field vanish anywhere? The space charge pattern will produce a field pattern given by Coulomb's law, and I do not see why or how that would vanish in any region.

---

## Round 4 · Referee Report · Anonymous (Referee 2) · 2023-5-8

Report

Having read the authors' reply, my impression is that the effect predicted by this paper is very difficult to be observed experimentally. The expected correlation between width and conductivity is expected to occur in a a very narrow temperature range (less than Tc/100). This requires an unrealistically high level of homogeneity and sharpness of transition combined with a miraculous shutdown of other contributions to conductivity. Because of this irrelevance to actual experiments, I do not find that the paper qualifies for publication in Scipost Physics.

---

## Round 4 · Author Response

Please see our response letter at: https://www.dropbox.com/s/8n1brebcxzxit1z/response_to_referee___superconductivity.pdf?dl=0.

(It would be nice if SciPost would add a button to attach PDF reply when resubmitting a manuscript :) ).

---

## Editorial Decision

awaiting_resubmission